# An Energy-Based Framework for Arbitrary Label Noise Correction

## Abstract

We propose an energy-based framework for correcting mislabelled training examples in the context of binary classification. While existing work addresses random and class-dependent label noise, we focus on feature dependent label noise, which is ubiquitous in real-world data and difficult to model. Two elements distinguish our approach from others: 1) instead of relying on the original feature space, we employ an autoencoder to learn a discriminative representation and 2) we introduce an energy-based formalism for the label correction problem. We prove that a discriminative representation can be learned by training a generative model using a loss function comprised of the difference of energies corresponding to each class. The learned energy value for each training instance is compared to the original training labels and contradictions between energy assignment and training label are used to correct labels. We validate our method across eight datasets, spanning synthetic and realistic settings, and demonstrate the technique's state-of-the-art label correction performance. Furthermore, we derive analytical expressions to show the effect of label noise on the gradients of empirical risk.

## 1 Introduction

Machine learning algorithms depend on reliable training labels or, when applicable, sufficiently robust learning models in order to produce generalizable predictions (Zhu & Wu, 2004; Frénay & Verleysen, 2014). Many empirical datasets suffer from training label corruption, which can result from annotation error, human bias, or a noisy process for generating labels (Smyth, 1996; Brodley & Friedl, 1999). In practice, it can be costly to obtain noise-free labels; hence, research on reducing the effects of label noise on learning has received considerable attention.

Most work, however, focuses on label noise that is independent of the input features (e.g., random label noise or class conditional random noise). The introduction of feature dependency significantly complicates mathematical analysis (Natarajan et al., 2013; Liu & Tao, 2014; Northcutt et al., 2017). Although work in this domain has yielded impressive results and improved generalization (Rebbapragada & Brodley, 2007a; Natarajan et al., 2013; Patrini et al., 2017; Rolnick et al., 2017; Ren et al., 2018), the simplicity of these noise assumptions fails to capture crucial mislabelling processes that arise in practice. In this work, we propose a semi-supervised framework for correcting the effects of feature dependent label noise on supervised learning algorithms.

Label noise processes are categorized into three types, as illustrated in Figure 1. Type I refers to a noise model where any label in the training data is incorrect with probability $\gamma$. In the case of type II noise, the probability of label corruption is conditioned on the class, such that we have different noise rate $\gamma_0$ and $\gamma_1$ for each class. Finally, type III noise models describe how label noise depends explicitly on input features. Feature dependent label noise or equivalently type III noise is a more realistic type of label noise that is ubiquitous in empirical datasets (Lachenbruch, 1974; Schafer & Graham, 2002; Frénay & Verleysen, 2014). As an example of type III noise, consider the diagnostic labels of Alzheimer's disease. In this setting, the probability of label noise depends on both age and sex (i.e., younger male patients are harder to diagnose) (Khachaturian, 1985; Murray et al., 2016). Type III noise also includes the prevalence of unreliable labels for training instances in low density regions of feature space (Denoeux, 1995; 1997; 2000) or near classification boundaries (Lachenbruch, 1966; 1974; Chhikara & McKeon, 1984; Cohen, 1997; Beigman & Klebanov, 2009; Beigman Klebanov & Beigman, 2009; Kolcz & Cormack, 2009).

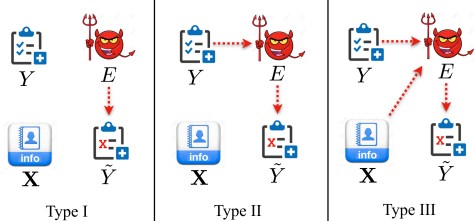

Figure 1: Different categories of label noise and their statistical dependencies, as depicted by the red arrow. In type I noise, all instances are equally likely to be mislabelled base on some probability $\gamma \in [0, \frac{1}{2}]$. In the case of type II, this probability is different for each class: $\gamma_0 \in [0, \frac{1}{2}]$ and $\gamma_1 \in [0, \frac{1}{2}]$. Type III label noise is the most realistic model and yet the least studied. In this case the probability of an error is a function of the input features: i.e. $p_{\text{error}} \sim f(\boldsymbol{x})$.

In this paper, we present a practical framework for correcting label noise that extends beyond type I and type II noise. We train an energy-based generative model on a small subset of the training data with reliable labels, either manually annotated or automatically learned. This model is used to identify and correct mislabelled examples based on the whether the assigned energy of a given point is compatible with its original label. We empirically demonstrate our framework's ability to handle input dependent label noise on both simulated and real datasets.

## 2 RELATED WORK

The full body of work on the problem of label noise is too extensive to review here; however, we direct interested readers to the detailed review by Frénay & Verleysen (2014). In this section, we highlight key contributions in the label noise literature. We then briefly discuss existing theoretical work.

### 2.1 LABEL NOISE CORRECTION

Existing work in label noise correction is designed for Type I and Type II noise and falls into three categories: relabeling, learning procedures and loss functions. We expand on each area below.

**Relabeling**: Earlier approaches to label noise focus on relabeling the noisy training set. Brodley & Friedl (1999) use the output of an ensemble of classifiers to identify mislabeled training examples. Sun et al. (2007) identify mislabeled instances based on the entropy of class probabilities outputted by a Bayesian classifier. These approaches, while robust to simpler label noise models, do not account for Type III noise.

**Learning Procedures**: Methods modifying the learning procedure include the perceptron algorithm with margin (PAM) Frénay & Verleysen (2014). Crammer & Lee (2010) use a velocity-based learning procedure to learn the weight vector distribution, termed gaussian herding (NHERD). Sukhbaatar et al. (2014) introduce a noise layer to a neural network architecture to learn the noise function, while Rebbapragada & Brodley (2007b) weight each example by class confidence in the training procedure. These methods commonly suffer from overfitting to the noise and again, primarily focus on Type I and Type II noise.

**Loss Functions**: Long & Servedio (2010) have shown classification algorithms that optimize a convex potential over a linear class are not robust to random label noise. This has led to work which aims to modify convex loss functions in order to make them noise-tolerant in the presence of type I and type II noise. Ghosh et al. (2015) derived sufficient conditions for classification losses that render them robust to random noise; namely, the components of a loss (where each component is defined over a given class) must sum to a constant value. Rooyen et al. (2015) proposed a convex loss that avoids the negative result of Long & Servedio (2010) for type I noise by virtue of being negatively unbounded. Natarajan et al. (2013) have shown that risk minimization over the corrupted data is consistent with risk minimization over clean data provided that the standard convex loss (e.g., binary cross entropy) is modified appropriately to produce an unbiased loss estimator (ULE). Each of these loss functions is presented for and validated on Type I and Type II noise.

## 2.2 EXISTING THEORETICAL GUARANTEES

(Bylander, 1994; Blum et al., 1998; Blum & Mitchell, 1998) offer guarantees for hypothesis generalization in the face of Type I and Type II noise. Angluin & Laird (1988); Bylander (1997; 1998); Servedio (1999) present guarantees for type III noise, where the probability of error depends on the distance to the margin. We introduce a gradient-based interpretation of label noise to these existing results.

## 3 BINARY CLASSIFICATION IN THE PRESENCE OF LABEL NOISE

### 3.1 PROBLEM SETUP AND NOTATION

Let $P$ denote the true distribution from which $n$ i.i.d. training examples $(\boldsymbol{x}_1, y_1), (\boldsymbol{x}_2, y_2), \ldots, (\boldsymbol{x}_n, y_n)$ have been drawn, where $(\boldsymbol{x}_i, y_i) \in \mathbb{R}^d \times \{0, 1\}$. The clean and corrupted training datasets are denoted as

$$\mathcal{T} = \big\{ (\boldsymbol{x}_i, y_i) \text{ for } i = 1, 2, \ldots, n \big\} \quad \& \quad \tilde{\mathcal{T}} = \big\{ (\boldsymbol{x}_i, \tilde{y}_i) \text{ for } i = 1, 2, \ldots, n \big\}, \tag{1}$$

respectively, where due to some label noise process $y_i \to \tilde{y}_i$. Hence, we have access to the noisy data $\tilde{\mathcal{T}}$ during training instead of the clean data $\mathcal{T}$. It can be assumed that the corrupted samples $(\boldsymbol{x}_i, \tilde{y}_i)$ are drawn from a noisy distribution $\tilde{P}$.

In supervised binary classification, we aim to learn a discriminator $f : \mathbb{R}^d \to \mathbb{R}$ that minimizes the *risk* with respect to a given loss function[1]. Formally, we want to minimize the empirical risk is $\hat{R}[\ell, f, \mathcal{T}] = \frac{1}{m} \sum_{i=1}^{m} \ell\left(f(\boldsymbol{x}_i, \boldsymbol{\theta}), y_i\right)$, where the $m$, $\ell$, $\boldsymbol{\theta}$ denote the size of the mini-batch size, loss function, and the learnable model parameters respectively. Gradient based learning algorithms compute $\nabla_{\boldsymbol{\theta}}(\hat{R}[\ell, f, \mathcal{T}])$ and update model parameters. For example, as in mini-batch gradient descent,

$$\boldsymbol{\theta}_{t+1} = \boldsymbol{\theta}_t - \eta \nabla_{\boldsymbol{\theta}} \left( \hat{R}[\ell, f, \mathcal{T}] \right), \tag{2}$$

where $\eta$ is the learning rate. In practice, we may not have access to the clean data $\mathcal{T}$. Instead we must learn the discriminator using the noisy data $\tilde{\mathcal{T}}$. Thus, the second term in Eq. 2 becomes $\nabla_{\boldsymbol{\theta}}(\hat{R}[\ell, f, \tilde{\mathcal{T}}])$, which denotes the risk with respect to the noisy training data.

### 3.2 LABEL NOISE EFFECT ON RISK MINIMIZATION

We can describe the effect of label noise on learning by expressing the noisy gradient $\nabla_{\boldsymbol{\theta}}(\hat{R}[\ell, f, \tilde{\mathcal{T}}])$ in terms of the true gradient $\nabla_{\boldsymbol{\theta}}(\hat{R}[\ell, f, \mathcal{T}])$ and a noise term. In the case of type I noise where any label $y_i$ can be flipped to $1 - y_i$ with probability $\gamma \in [0, 1/2]$, we have

$$\mathbb{E}_{(\boldsymbol{x}, \tilde{y}) \sim \tilde{P}} \left[ \nabla_{\boldsymbol{\theta}} \hat{R}[\ell, \tilde{\mathcal{T}}] \right] = (1 - 2\gamma) \, \mathbb{E}_{\text{true}} + 2\gamma \, \mathbb{E}_{\text{rand}}, \tag{3}$$

where $\mathbb{E}_{\text{true}} \equiv \mathbb{E}_{(\boldsymbol{x}, \tilde{y}) \sim \tilde{P}}[\nabla_{\boldsymbol{\theta}} \hat{R}[\ell, \mathcal{T}]]$ is the true (i.e. noise-free) gradient, and $\mathbb{E}_{\text{rand}}$ is the completely noisy term obtained by replacing all targets $y_i$ with a random value of either 0 or 1. For type II noise, we have an equation analogous to Eq. 3, where $\gamma$ is replaced by the average $\bar{\gamma} = (\gamma_0 + \gamma_1)/2$ noise rate and we have an additional term reflecting the noise imbalance between each class when $\gamma_0 \neq \gamma_1$. For more information on this and derivations of these equations, see Appendix A.

From Eq. 3 we deduce that for small $\gamma$ the true gradients dominate the dynamics of searching $\boldsymbol{\theta}$ space for optimal parameters. As $\gamma$ grows, the true gradients are scaled down by the factor $(1 - 2\gamma)$, which means that the contribution of $\mathbb{E}_{\text{true}}$ diminishes and the random perturbation term $\mathbb{E}_{\text{rand}}$ starts to dominate as $\gamma$ grows. A similar conclusion can be reached for type II noise (replacing $\gamma$ with $\bar{\gamma}$) where we have an additional term modifying the true gradients based on the class imbalance (see A.2).

The effect of $\mathbb{E}_{\text{rand}}$ can be negated by increasing the amount of training data proportionately with $\bar{\gamma}$ as discussed by Rolnick et al. (2017): i.e. as the probability of a random error occurring increases,

---

[1] In practice the loss function $\ell$ is usually *classification-calibrated* to minimize the 0-1 loss given that the sample size is sufficiently large (Bartlett et al., 2006).

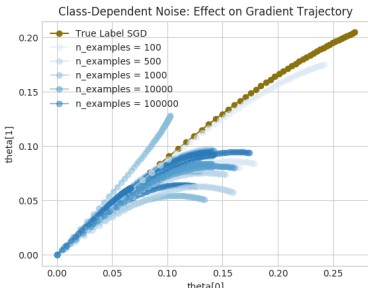 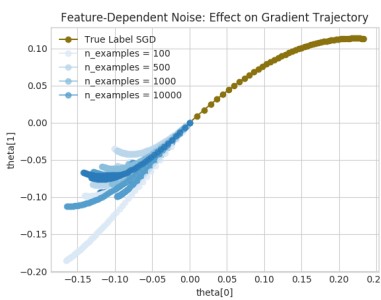

Figure 2: Plots showing the path traversed in $\boldsymbol{\theta}$ space by a supervised model as learnable parameters are updated via a gradient based optimizer. The gold line shows the path followed by a noise-free model and the blue lines show the modified paths due to label noise. In the case of random label noise (left plot), the perturbations $\mathbb{E}_{\text{rand}}$ to the true gradient direction are curbed by an increase in training set size. In contrast, for input dependent noise (right plot), supplying the model with more data perpetuates the label bias.

the model needs more data to make progress in the direction of the true gradient. We can neutralize the effects (i.e. suppress $\mathbb{E}_{\text{rand}}$) of random noise (type I) or class conditional random noise (type II) by increasing the number of training examples and by utilizing a robust loss function when possible (Natarajan et al., 2013).

In contrast, Type III noise is substantially more difficult to address because the available training data $\tilde{\mathcal{T}}$ can have optimal discriminators that are significantly different from the true discriminator corresponding to the clean data $\mathcal{T}$. Separating Type III noise into the true term and the perturbation term necessitates restrictive assumptions on the noise model (See Appendix A. In Figure 2, we show this by visualizing SGD over the true training set (gold lines) and the noise training set (blue lines). As shown in Figure 2, increasing the size of the training set is effective in address Type II noise. This strategy is not only ineffective for addressing type III noise, but they can also lead to our model learning the biased discriminator; thus, hindering any possibility of generalization.

## 4  AN ENERGY-BASED FRAMEWORK FOR BIAS CORRECTION

Type III noise is particularly challenging because noise can depend on the input features in an arbitrarily complex way. Modeling this requires significant information about the noise process, which can be difficult to obtain in practice. In order to deal with this challenge we will avoid training directly on the mislabelled instances. Instead, we start our training procedure by assuming there exists a small subset of $\tilde{\mathcal{T}}$ that is noise-free (e.g. $1\%$ of the data is sufficient the MNIST dataset). Such a set can be obtained via a manual labelling process or by using domain knowledge to identify the appropriate data points[2].

Our objective is to correct mislabelled training instances such that the corrected data leads to the improved generalization of a trained model. We will demonstrate in Section 5 that our framework extends beyond Type I and Type II noise to Type III noise, in realistic settings. In particular, we demonstrate the method's strength in correcting algorithmically assigned labels.

The proposed correction framework consists of three steps:

1. *Obtain known labels*: There are two ways of identifying instances that have correct labels. The first approach is to use domain knowledge – frequently, one has access to a subset of clean labels and their instances. We can use this subset as our clean data for step 2. If this set is unknown or too small, then we can inference which examples are likely to be correctly labeled, as is done in Ding et al. (2018). We report results on both approaches in Section 5.

---

[2]This assumption can be relaxed in situations where we can learn which labels are likely to be clean. E.g. see Ding et al. (2018)

2. *Train semi-supervised model*: The framework uses the filtered data from the previous step to train a semi-supervised algorithm that learns to classify instances based on similarity between features. In contrast with supervised learning, where the goal is to learn $p(y|\boldsymbol{x})$, we want to learn which features are most compatible with a given label: i.e. $p(\boldsymbol{x}|y)$. Thus, we train a generative model (e.g. energy-based autoencoder) to use feature similarity (as quantified by an energy function) to identify which instances belong to the specific class.

3. *Identify and correct mislabelled instances*: The features of the full training data (with mislabeled targets) are injected into the previously trained semi-supervised model resulting in each instance being assigned an energy value based on the output of the model. The energy corresponding to each instance serves as a proxy for class assignment (i.e. low energy corresponds to $y = 0$ whereas high energy corresponds to $y = 1$). Contradictions between energy assignment and training labels are used to correct the training data such that all labels are compatible with their assigned energy.

We first motivate the design decisions involved in this framework. Namely, we explore our choice of contrastive divergence as a loss function and empirically justify our use of an autoencoder. In the subsequent section, we elaborate on our implementation of the framework.

## 4.1 FRAMEWORK MOTIVATION

Given a small, clean dataset $\mathscr{T} \subset \tilde{\mathcal{T}}$, we aim to train a model which can be used to determine labels on the remaining, noisily labelled dataset. One obvious approach is to train a binary classifier directly and use it to correct labels. However, as discussed by Berthelot et al. (2017), binary classification provides a relatively weak training signal which largely ignores the intricacies of the input feature distribution. See Fig. 3 where we compare the class separation achieved by a few standard binary classifiers with our proposed approach. Alternatively, one could train class conditional models. However, such models are trained without knowledge of the primary task, differentiating classes. Further, traditional unsupervised training regimes typically use a maximum likelihood formulation (i.e. forward KL divergence) which is prone to distributing probability mass broadly (the so-called "mean-seeking behaviour" (Murphy, 2012)) and further limit the discriminative ability of the learned model.

Instead, we prioritize learning to discriminate the underlying classes based on aspects of the input feature distributions. To do this, we propose to use a contrastive divergence (Carreira-Perpinan & Hinton, 2005) training loss:

$$\text{CD}\left(\boldsymbol{\theta}\right) = \text{KL}\left[p_0\left(\boldsymbol{x}\right) \| q\left(\boldsymbol{x}; \boldsymbol{\theta}\right)\right] - \text{KL}\left[p_1\left(\boldsymbol{x}\right) \| q\left(\boldsymbol{x}; \boldsymbol{\theta}\right)\right]. \tag{4}$$

where $p_0(\boldsymbol{x})$ and $p_1(\boldsymbol{x})$ denote the target probability distributions conditioned on class 0 and 1 respectively and $q(\boldsymbol{x}; \theta)$ denotes the learned model. Thus, training aims to produce a model $q\left(\boldsymbol{x}; \boldsymbol{\theta}\right)$ which assigns high probability to samples from class 0, while giving low probability to samples from class 1. In theorem 1, we show that a finite sample approximation of the optimization objective in Eq. 4 can be computed as

$$\mathcal{L}\left(\boldsymbol{\theta}\right) = \frac{1}{|\mathscr{T}_0|} \sum_{\boldsymbol{x}_0 \in \mathscr{T}_0} E\left(\boldsymbol{x}_0; \boldsymbol{\theta}\right) - \frac{1}{|\mathscr{T}_1|} \sum_{\boldsymbol{x}_1 \in \mathscr{T}_1} E\left(\boldsymbol{x}_1; \boldsymbol{\theta}\right), \tag{5}$$

where $\mathscr{T}_y$, for $y \in \{0, 1\}$, contains instances from class $y$ and $\mathscr{T}_y \subseteq \mathscr{T}$, $|\mathscr{T}_y|$ denotes the number of elements in $\mathscr{T}_y$ and $E(\boldsymbol{x}; \boldsymbol{\theta})$ is the energy of the model $q\left(\boldsymbol{x}; \boldsymbol{\theta}\right)$, i.e., $q\left(\boldsymbol{x}; \boldsymbol{\theta}\right) = \exp\left(-E\left(\boldsymbol{x}; \boldsymbol{\theta}\right)\right)/Z(\boldsymbol{\theta})$ where $Z(\boldsymbol{\theta})$ is the normalizing partition function. Analogously, we can define Eq. 5 over each mini-batch. This contrastive loss have been applied successfully in adversarial training, resulting in faster and improved stability of training, enhanced generator quality, and improved generator diversity (Zhao et al., 2016; Berthelot et al., 2017). As described in section 2.2 of (LeCun et al., 2006), the negative log-likelihood function, $E(\boldsymbol{x}; \boldsymbol{\theta}) = -\log q\left(\boldsymbol{x}; \boldsymbol{\theta}\right)$, constitute a valid energy which has been used in numerous energy-based models [e.g. (Bengio et al., 2003; Zhao et al., 2016; Berthelot et al., 2017)].

**Theorem 1.** *The contrastive divergence* $\text{CD}\left(\boldsymbol{\theta}\right)$ *in Equation 4 has the finite approximation* $\mathcal{L}\left(\boldsymbol{\theta}\right)$ *up to some constant k which is independent of* $\boldsymbol{\theta}$.

*Proof.* Using the definition of KL divergence, we expand Equation 4:

$$\text{CD}\left(\boldsymbol{\theta}\right) = -\int_{\boldsymbol{x}} p_0\left(\boldsymbol{x}\right) \log q\left(\boldsymbol{x}; \boldsymbol{\theta}\right) d\boldsymbol{x} + \int_{\boldsymbol{x}'} p_1\left(\boldsymbol{x}'\right) \log q\left(\boldsymbol{x}'; \boldsymbol{\theta}\right) d\boldsymbol{x}' + k \tag{6}$$

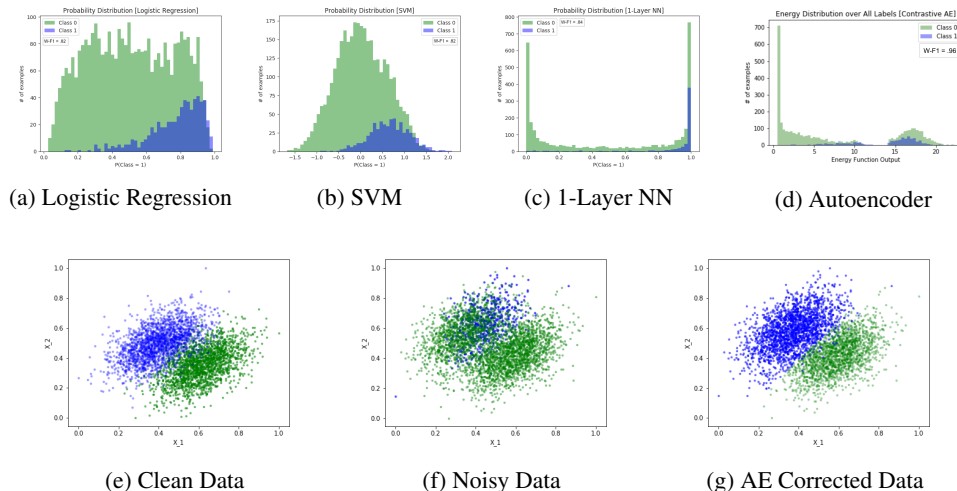

Figure 3: In the first row, we visualize the separation achieved by classifiers trained on a training set with noisy labels. From left to right, we show the results of logistic regression, an SVM, a 1 layer feed-forward neural network and the proposed technique. In the second row, we visualize the training data. From left to right, we show the clean data, the noisy data and the autoencoder-corrected data. We observe that standard binary classifiers are unable to achieve satisfactory class separation. The 1 hidden layer NN classifier accomplishes significant separation but it incorrectly identifies mislabelled instances. The proposed model is able to identify mislabelled instances based on the energy distribution of the training data.

where the two entropy terms $\int_{\boldsymbol{x}} p_0\left(\boldsymbol{x}\right)\log p_0\left(\boldsymbol{x}\right)d\boldsymbol{x}$ and $\int_{\boldsymbol{x}} p_1\left(\boldsymbol{x}\right)\log p_1\left(\boldsymbol{x}\right)d\boldsymbol{x}$ have been grouped in the constant $k$ as they do not depend on the optimization parameters $\boldsymbol{\theta}$. Assume the Gibbs form for the model distribution,

$$q\left(\boldsymbol{x};\boldsymbol{\theta}\right) = \frac{e^{-E(\boldsymbol{x};\boldsymbol{\theta})}}{Z(\boldsymbol{\theta})}, \tag{7}$$

where $E(\boldsymbol{x};\boldsymbol{\theta})$ is the energy and $Z(\boldsymbol{\theta}) = \int_{\boldsymbol{x}} e^{-E(\boldsymbol{x};\boldsymbol{\theta})}d\boldsymbol{x}$ is the partition function. Then, Equation 6 becomes

$$\text{CD}\left(\boldsymbol{\theta}\right) = \int_{\boldsymbol{x}} p_0\left(\boldsymbol{x}\right)E(\boldsymbol{x};\boldsymbol{\theta})d\boldsymbol{x} - \int_{\boldsymbol{x'}} p_1\left(\boldsymbol{x'}\right)E(\boldsymbol{x};\boldsymbol{\theta})d\boldsymbol{x'} \tag{8}$$

$$- \left(\int_{\boldsymbol{x}} p_0\left(\boldsymbol{x}\right)d\boldsymbol{x}\right)Z(\boldsymbol{\theta}) + \left(\int_{\boldsymbol{x}} p_1\left(\boldsymbol{x}\right)d\boldsymbol{x}\right)Z(\boldsymbol{\theta}) + k, \tag{9}$$

where the partition function terms conveniently cancel out due to the normalization of $p_0$ and $p_1$. Therefore,

$$\text{CD}\left(\boldsymbol{\theta}\right) = \mathbb{E}_{p_0(\boldsymbol{x})}\left[E\left(\boldsymbol{x}\right)\right] - \mathbb{E}_{p_1(\boldsymbol{x})}\left[E\left(\boldsymbol{x}\right)\right] + k \tag{10}$$

and we conclude that $\text{CD}\left(\boldsymbol{\theta}\right)$ has the finite approximation $\mathcal{L}\left(\boldsymbol{\theta}\right)$ as defined in Equation 5, up to a constant $k$. $\qquad\square$

## 4.2 FRAMEWORK IMPLEMENTATION: CONTRASTIVE AUTOENCODER

Using the clean subset $\mathscr{T}$, we implement step 2 of our framework by using an energy-based autoencoder (analogous to Zhao et al. (2016)). In order to ensure stable training and to prevent the second term of Equation 5 from diverging, we balance our mini-batch to have an equal number of positive and negative class samples. After training is complete, we process the original training data $\tilde{\mathcal{T}}$ using the trained autoencoder. Specifically, we compute the energy using the resulting outputs: $E\left(\boldsymbol{x};\boldsymbol{\theta}\right) = -\log q\left(\boldsymbol{x};\boldsymbol{\theta}\right)$, which reduces to the reconstruction loss corresponding to the autoencoder (Goodfellow et al., 2016). For example, if the underlying distribution is assumed to be Gaussian then, we have $E\left(\boldsymbol{x};\boldsymbol{\theta}\right) = \|\boldsymbol{x} - \hat{\boldsymbol{x}}(\boldsymbol{\theta})\|^2$, where $\hat{\boldsymbol{x}}(\boldsymbol{\theta})$ is the output of

the autoencoder[3]. Whereas, if the underlying distribution is assumed to be Bernoulli, we obtain $E\left(\boldsymbol{x} ; \boldsymbol{\theta}\right) = -\left[\boldsymbol{x} \log \hat{\boldsymbol{x}}(\boldsymbol{\theta}) + (1 - \boldsymbol{x}) \log(1 - \hat{\boldsymbol{x}}(\boldsymbol{\theta}))\right]$.

To execute our label correction protocol, we impose the following two conditions. (i) if $E(\boldsymbol{x} ; \boldsymbol{\theta}) > \bar{E}_1 - a$ for $\boldsymbol{x} \in \tilde{\mathcal{T}}_0$, then change label of $\boldsymbol{x}$. Here we denote $\tilde{\mathcal{T}}_0$ as the originals negative samples, $\bar{E}_1$ as the mean energy over the set of clean positive samples $\mathscr{T}_1$, and $a$ is a tuneable hyperparameter that determines how aggressively we want to change $0 \rightarrow 1$. Analogously, we impose condition (ii) if $E(\boldsymbol{x} ; \boldsymbol{\theta}) < \bar{E}_0 - b$ for $\boldsymbol{x} \in \tilde{\mathcal{T}}_1$, then change label of $\boldsymbol{x}$. The rationale is to modify samples where the original label $\tilde{y}$ contradicts the energy assignment. When one is not able to tune the threshold hyperparameters $a$ and $b$, an empirical heuristic is to set them equal to the standard deviation about the means $\bar{E}_1$ and $\bar{E}_0$ respectively. See Appendix B details and pseudo-code.

## 5 EXPERIMENTS

We conduct experiments with simulated data and real-world data where three different type III noise models are studied: (i) *Linear noise* ($p_{\text{error}} \sim \alpha x_i$): the probability of an error occurring depends linearly on a feature (Table 1). (ii) *Quadratic noise* ($p_{\text{error}} \sim \alpha x_i^2$): the probability of an error occurring depends on the square of a feature. We control the amount of noise added with $\alpha$ and we select $x_i$ if it has sufficient variance to make the noise model distinct from random noise, i.e. $0$ variance implies random noise (Table 2). (iii) *Boundary noise*: the probability of label error depends on the distance from the class boundary, which is determined using the noise-free data. We report the average class-weighted F1-score along with the standard error over 10 runs with random train-test (80:20) splits (Table 3).

**UCI benchmark datasets**: We train logistic regression on the corrected dataset resulting from the proposed algorithm. If a small clean dataset is not initially provided, then our results are labelled as AE (learned), otherwise they are labelled at AE (known) – see step 1 in Sec. 4. We compare to noise robust algorithms: NHERD Crammer & Lee (2010), PAM Frénay & Verleysen (2014), and ULE Natarajan et al. (2013). Additionally, we report upper and lower bounds by training logistic regression on the noisy data (LR-N) and on the clean data (LR-C). All algorithms are tested on noise-free data. Even without access to any clean labels (i.e., we learn which labels are likely clean), our framework generally outperforms the other algorithms in the presence of linear noise and quadratic noise. In general, learning which labels are noisy following the method in Ding et al. (2018) is not possible for arbitrary label noise processes: e.g., when the optimal discriminators of $\tilde{\mathcal{T}}$ are very different than the optimal discriminators of $\mathcal{T}$. For boundary noise, we show that using a small clean dataset $\mathscr{T}$ enables our methods to learn in the presence of boundary noise, where the other methods generally breakdown (Table 3). For a fair evaluation, the benchmark algorithms (which don't have access to clean data) should be compared to AE (learned).

**MNIST**: Next, we compare our method with another state-of-the-art noise robust algorithm that relies on clean data, i.e., the loss weighting scheme of Ren et al. (2018). Both methods are given the same percentage of clean data for each noise setting. The task is to classify the distinguish "3" (class 0) vs. other digits (class 1). The noise process is input dependent as it changes $4, 5, 6$ to class 0 depending on a specified noise rate ($\alpha$). We demonstrate (Table 4) that our method generalizes better in the presence of type III noise, even when starting with only $1\%$ of the data that is clean.

**Arrhythmia** We use the MIT-BIH Arrhythmia dataset Goldberger et al. (2000) to evaluate the proposed method's ability to correct algorithmically assigned labels. Algorithmically-assigned labels (AALs) are prevalent in domains with abundant unlabeled data and high labelling costs. Common applications include web page annotation, but the value of this approach extends to automatic annotation of images and natural language. We employ the data preprocessing procedure described by Mondéjar-Guerra et al. (2019), where each arrhythmia is described by 59 features.

We divide the dataset into three parts: $T_1$, $T_2$ and $T_3$. $T_1$ and $T_2$ serve as training sets and $T_3$ is the test set. We train a classifier on $T_1$ and subsequently use that classifier to generate AALs for $T_2$. Thus, the predictions of the trained classifier become $\tilde{y}$ and the original expert labels remain $y$. As demonstrated in Table 1, the proposed technique achieves a higher F1 score than competing

---

[3]More precisely, $E\left(\boldsymbol{x} ; \boldsymbol{\theta}\right) = \|\boldsymbol{x} - \hat{\boldsymbol{x}}(\boldsymbol{\theta})\|^2$ up to an irrelevant scaling factor that we can drop for simplicity.

| Dataset | Noise Parameters (col, $\alpha$) | LR-N | NHERD | PAM | ULE | AE (learned) | AE (known) | LR-C |
|---|---|---|---|---|---|---|---|---|
| Lin-Sep | 1 , 1.2 | $0.86 \pm 0.03$ | $0.91 \pm 0.01$ | $0.76 \pm 0.01$ | $0.90 \pm 0.02$ | $0.94 \pm 0.01$ | $0.95 \pm 0.01$ | $0.96 \pm 0.01$ |
| Diabetes | 5 , 1.0 | $0.60 \pm 0.03$ | $0.50 \pm 0.01$ | $0.48 \pm 0.03$ | $0.60 \pm 0.03$ | $0.64 \pm 0.01$ | $0.70 \pm 0.02$ | $0.72 \pm 0.02$ |
| German | 1 , 1.2 | $0.67 \pm 0.03$ | $0.72 \pm 0.01$ | $0.49 \pm 0.03$ | $0.63 \pm 0.01$ | $0.67 \pm 0.03$ | $0.73 \pm 0.01$ | $0.76 \pm 0.02$ |
| Image | 1 , 0.7 | $0.61 \pm 0.03$ | $0.63 \pm 0.01$ | $0.54 \pm 0.01$ | $0.61 \pm 0.02$ | $0.67 \pm 0.01$ | $0.77 \pm 0.01$ | $0.77 \pm 0.01$ |
| Twonorm | 1 , 1.2 | $0.68 \pm 0.04$ | $0.50 \pm 0.02$ | $0.43 \pm 0.03$ | $0.82 \pm 0.04$ | $0.88 \pm 0.01$ | $0.91 \pm 0.02$ | $0.98 \pm 0.01$ |
| Breast Cancer | 5 , 1.0 | $0.66 \pm 0.02$ | $0.67 \pm 0.02$ | $0.52 \pm 0.03$ | $0.60 \pm 0.03$ | $0.60 \pm 0.02$ | $0.71 \pm 0.02$ | $0.70 \pm 0.01$ |
| Arrhythmia | - | $0.79 \pm 0.01$ | $0.83 \pm 0.01$ | $0.81 \pm 0.02$ | $0.66 \pm 0.04$ | $0.85 \pm 0.02$ | $0.85 \pm .01$ | $0.85 \pm 0.02$ |

Table 1: Noise model: probability of an error occurring depends linearly on an input feature. We report the class weighted mean f1 score on the noise-free test set along with the standard error.

| Dataset | Noise Parameters (col, $\alpha$) | LR-N | NHERD | PAM | ULE | AE (learned) | AE (known) | LR-C |
|---|---|---|---|---|---|---|---|---|
| Lin-Sep | 1 , 1.2 | $0.40 \pm 0.01$ | $0.53 \pm 0.04$ | $0.62 \pm 0.03$ | $0.61 \pm 0.02$ | $0.93 \pm 0.01$ | $0.96 \pm 0.01$ | $0.96 \pm 0.01$ |
| Diabetes | 5 , 1.2 | $0.62 \pm 0.01$ | $0.59 \pm 0.01$ | $0.67 \pm 0.01$ | $0.71 \pm 0.01$ | $0.66 \pm 0.03$ | $0.69 \pm 0.01$ | $0.72 \pm 0.02$ |
| German | 1 , 1.2 | $0.60 \pm 0.02$ | $0.72 \pm 0.01$ | $0.60 \pm 0.01$ | $0.67 \pm 0.01$ | $0.68 \pm 0.03$ | $0.73 \pm 0.03$ | $0.76 \pm 0.02$ |
| Image | 1 , 0.7 | $0.55 \pm 0.02$ | $0.64 \pm 0.01$ | $0.57 \pm 0.01$ | $0.60 \pm 0.03$ | $0.74 \pm 0.01$ | $0.77 \pm 0.01$ | $0.77 \pm 0.01$ |
| Twonorm | 1 , 1.2 | $0.90 \pm 0.03$ | $0.55 \pm 0.02$ | $0.86 \pm 0.01$ | $0.94 \pm 0.01$ | $0.96 \pm 0.01$ | $0.97 \pm 0.01$ | $0.98 \pm 0.01$ |
| Breast Cancer | 5 , 1.0 | $0.60 \pm 0.02$ | $0.60 \pm 0.02$ | $0.63 \pm 0.02$ | $0.63 \pm 0.03$ | $0.65 \pm 0.03$ | $0.66 \pm 0.01$ | $0.70 \pm 0.01$ |

Table 2: Noise model: probability of an error occurring depends quadratically on an input feature. We report the class weighted mean f1 score on the noise-free test set along with the standard error.

| Dataset | Noise Parameters ($\alpha$) | LR-N | NHERD | PAM | ULE | AE (known) | LR-C |
|---|---|---|---|---|---|---|---|
| Lin-Sep | 0.7 | $0.39 \pm 0.01$ | $0.41 \pm 0.01$ | $0.53 \pm 0.02$ | $0.85 \pm 0.01$ | $0.94 \pm 0.01$ | $0.96 \pm 0.01$ |
| Diabetes | 0.7 | $0.56 \pm 0.01$ | $0.53 \pm 0.03$ | $0.50 \pm 0.02$ | $0.55 \pm 0.02$ | $0.68 \pm 0.02$ | $0.72 \pm 0.02$ |
| German | 0.7 | $0.57 \pm 0.01$ | $0.70 \pm 0.01$ | $0.47 \pm 0.01$ | $0.60 \pm 0.01$ | $0.72 \pm 0.01$ | $0.76 \pm 0.02$ |
| Image | 0.7 | $0.43 \pm 0.01$ | $0.61 \pm 0.01$ | $0.45 \pm 0.02$ | $0.43 \pm 0.01$ | $0.74 \pm 0.03$ | $0.77 \pm 0.01$ |
| Twonorm | 0.5 | $0.52 \pm 0.03$ | $0.51 \pm 0.03$ | $0.52 \pm 0.03$ | $0.51 \pm 0.04$ | $0.88 \pm 0.03$ | $0.98 \pm 0.01$ |
| Breast Cancer | 0.7 | $0.62 \pm 0.02$ | $0.62 \pm 0.02$ | $0.46 \pm 0.02$ | $0.63 \pm 0.02$ | $0.66 \pm 0.04$ | $0.70 \pm 0.01$ |

Table 3: Noise model: probability of an error occurring depends linearly on the distance from class boundary (which is defined by the clean dataset before noise is synthetically introduced). We report the class weighted mean f1 score on the noise-free test set along with the standard error.

| % Clean data | Noise Parameter ($\alpha$) | AE(known) | Ren et al. (2018) | No noise model |
|---|---|---|---|---|
| 1 | 0.1 | $98.28 \pm 0.01$ | $93.27 \pm 0.01$ | $90.87 \pm 0.05$ |
| 1 | 0.8 | $98.14 \pm 0.00$ | $91.27 \pm 0.01$ | $87.32 \pm 0.03$ |
| 5 | 0.1 | $98.77 \pm 0.00$ | $93.44 \pm 0.01$ | $91.01 \pm 0.02$ |
| 5 | 0.8 | $98.65 \pm 0.00$ | $91.29 \pm 0.01$ | $88.85 \pm 0.02$ |
| 10 | 0.1 | $98.93 \pm 0.00$ | $94.87 \pm 0.03$ | $90.99 \pm 0.02$ |
| 10 | 0.8 | $98.54 \pm 0.01$ | $91.77 \pm 0.02$ | $90.16 \pm 0.03$ |

Table 4: We compare the label re-weighting scheme of Ren et al. (2018) with our proposed model. The models compared for different sizes of clean dataset and different noise rates $\alpha$. The base model in each case is a standard LeNet as in Ren et al. (2018)

methods. The success of the model in this setting suggests its unique robustness to feature-dependent noise.

## 6 CONCLUSION

We have proposed an energy-based framework to correct mislabelled training instances. By minimizing a contrastive loss, the proposed method learns a representation that is valuable in relabeling noisy training sets. We evaluate the proposed model across six datasets and three noise models to demonstrate the method's empirical value in correcting feature-dependent label noise. Furthermore, we demonstrate our method's improvement upon existing work in making machine learning more robust to label noise processes.

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

APPENDICES

## A EFFECT OF LABEL NOISE ON EMPIRICAL RISK

Here we show how label noise affects optimization of empirical risk. Recall that our task is to improve the labels of a training set in order to enable effective learning. In the following sections, we explore the Type I, Type II, and Type III noise. We first introduce the key variables.

We focus on binary classification, where the probability of class 0 is $p$ and consequently, the probability of class 1 is $(1 - p)$:

$$
\begin{aligned}
p(y = 0) &= p \\
p(y = 1) &= 1 - p
\end{aligned}
$$

$\mathcal{T}$ represents the training set with correct labels, while $\tilde{\mathcal{T}}$ represents the training set with label noise. In our setting, we are given $\tilde{\mathcal{T}}$ and propose a method to transform $\tilde{\mathcal{T}}$ into $\mathcal{T}$. We will call $D$ the true label distribution and $\tilde{P}$ the noisy label distribution.

$$
\begin{aligned}
\mathcal{T} &= \big\{ (\mathbf{X}_i, Y_i) \text{ for } i = 1, 2, \ldots, n \big\} \\
\tilde{\mathcal{T}} &= \big\{ (\mathbf{X}_i, \tilde{Y}_i) \text{ for } i = 1, 2, \ldots, n \big\}
\end{aligned}
$$

In the following calculations, we describe the difference in empirical risks over each training set and the subsequent effect on stochastic gradient descent. We refer to the empirical risk over the clean training set as $\hat{R}[\ell, \mathcal{T}]$. Similarly, we refer to the empirical risk over the corrupted training set as $\hat{R}\left[\ell, \tilde{\mathcal{T}}\right]$.

$$
\hat{R}[\ell, \mathcal{T}] = \frac{1}{n} \sum_{i=1}^{n} \ell\left(\Phi(\mathbf{X}_i, \boldsymbol{\theta}), Y_i\right),
$$

In order to explain the effects of label noise on stochastic gradient descent (SGD), we are primarily interested in the gradient of the empirical risk with respect to $\theta$. Below, we break the empirical risk term into two class-dependent terms and introduce short-hand to represent each of these terms.

$$
\begin{aligned}
\mathbb{E}_{(\mathbf{X},Y)\sim D}\left[\nabla_{\boldsymbol{\theta}}\hat{R}[\ell, \mathcal{T}]\right] &= \mathbb{E}_{(\mathbf{X},Y)\sim D}\left[\nabla_{\boldsymbol{\theta}} \frac{1}{n} \sum_{i=1}^{n} \ell\left(\Phi(\mathbf{X}_i, \boldsymbol{\theta}), Y_i\right)\right] && (11) \\
&= \nabla_{\boldsymbol{\theta}} \frac{1}{n} \sum_{i=1}^{n} \mathbb{E}_{(\mathbf{X},Y)\sim D}\left[\ell\left(\Phi(\mathbf{X}_i, \boldsymbol{\theta}), Y_i\right)\right] && (12) \\
&= \nabla_{\boldsymbol{\theta}} \left[\frac{1}{n} \sum_{i=1}^{n} \int p(\mathbf{x}_i, y_i)\ell\left(\Phi(\mathbf{x}_i, \boldsymbol{\theta}), y_i\right) \, d\mathbf{x}_i \, dy_i\right] && (13) \\
&= \nabla_{\boldsymbol{\theta}} \left[\frac{1}{n} \sum_{i=1}^{n} \int p(\mathbf{x}_i|y_i)p(y_i)\ell\left(\Phi(\mathbf{x}_i, \boldsymbol{\theta}), y_i\right) \, d\mathbf{x}_i \, dy_i\right] && (14) \\
&= p(y_i = 0)\nabla_{\boldsymbol{\theta}} \left[\frac{1}{n} \sum_{i=1}^{n} \int p(\mathbf{x}_i|0)\ell_0\left(\Phi(\mathbf{x}_i, \boldsymbol{\theta})\right) \, d\mathbf{x}_i\right] && (15) \\
&\quad + p(y_i = 1)\nabla_{\boldsymbol{\theta}} \left[\frac{1}{n} \sum_{i=1}^{n} \int p(\mathbf{x}_i|1)\ell_1\left(\Phi(\mathbf{x}_i, \boldsymbol{\theta})\right) \, d\mathbf{x}_i\right],
\end{aligned}
$$

We define shorthand based on Equation (5):

$$\boldsymbol{\beta}_0 = \nabla_{\boldsymbol{\theta}} \left[ \frac{1}{n} \sum_{i=1}^{n} \int p(\mathbf{x}_i | 0) \ell_0 \left( \Phi(\mathbf{x}_i, \boldsymbol{\theta}) \right) \, d\mathbf{x}_i \right] \tag{16}$$

$$\boldsymbol{\beta}_1 = \nabla_{\boldsymbol{\theta}} \left[ \frac{1}{n} \sum_{i=1}^{n} \int p(\mathbf{x}_i | 1) \ell_1 \left( \Phi(\mathbf{x}_i, \boldsymbol{\theta}) \right) \, d\mathbf{x}_i \right] . \tag{17}$$

Rewriting the original equation in terms of the above, we arrive at the following form for empirical risk:

$$\mathbb{E}_{(\mathbf{X}, Y) \sim D} \left[ \nabla_{\boldsymbol{\theta}} \hat{R}[\ell, \mathcal{T}] \right] = p(y_i = 0) \boldsymbol{\beta}_0 + p(y_i = 1) \boldsymbol{\beta}_0 \tag{18}$$

We revisit Equation 8 in the follow sections to interpret empirical risk under varying noise models. Note that $\boldsymbol{\beta}_0$ and $\boldsymbol{\beta}_1$ are the same in both the expected empirical risk with respect to the true label distribution ($D$) and the corrupted label distribution ($\tilde{P}$) for Type I and Type II noise. Additionally, we introduce shorthand for the true expectation of empirical risk ($\mathbb{E}_{true}$) and the expectation of empirical risk when labels are assigned at random ($\mathbb{E}_{rand}$).

$$\mathbb{E}_{true} = p\boldsymbol{\beta}_0 + (1 - p)\boldsymbol{\beta}_1 \tag{19}$$
$$\mathbb{E}_{rand} = .5\boldsymbol{\beta}_0 + .5\boldsymbol{\beta}_1 \tag{20}$$

### A.1 Type I: Random Noise

In the random noise scenario, each label has an equal probability of being flipped. We parameterize this noise by $\gamma$, resulting in :

$$\gamma = p(\tilde{y}_i \neq y_i) \tag{21}$$

We may define both $p(\tilde{y}_i = 0)$ in terms of $\gamma$ and $p$:

$$p(\tilde{y}_i = 0) = p(\tilde{y}_i = 0 | y_i = 0) + p(\tilde{y}_i = 0 | y_i = 1) \tag{22}$$
$$p(\tilde{y}_i = 0) = (1 - \gamma)p + \gamma(1 - p) \tag{23}$$
$$\tag{24}$$

Likewise, we define $p(\tilde{y}_i = 1)$ as:

$$p(\tilde{y}_i = 1) = p(\tilde{y}_i = 1 | y_i = 0) + p(\tilde{y}_i = 1 | y_i = 1) \tag{25}$$
$$p(\tilde{y}_i = 1) = \gamma p + (1 - \gamma)(1 - p) \tag{26}$$

Below, we substitute these expressions into Equation 8:

$$
\begin{aligned}
\mathbb{E}_{(\mathbf{X}, \tilde{Y}) \sim \tilde{P}} \left[ \nabla_{\boldsymbol{\theta}} \hat{R}[\ell, \tilde{\mathcal{T}}] \right] &= p(\tilde{y}_i = 0)\boldsymbol{\beta}_0 + p(\tilde{y}_i = 1)\boldsymbol{\beta}_0 & (27) \\
&= \left( (1 - \gamma) p + \gamma - \gamma p \right) \boldsymbol{\beta}_0 + \left( (1 - \gamma)(1 - p) + \gamma p \right) \boldsymbol{\beta}_1 & (28) \\
&= p\boldsymbol{\beta}_0 - 2\gamma p\boldsymbol{\beta}_0 + \gamma\boldsymbol{\beta}_0 + \boldsymbol{\beta}_1 - \gamma\boldsymbol{\beta}_1 - p\boldsymbol{\beta}_1 + 2\gamma p\boldsymbol{\beta}_1 & (29) \\
&= p\boldsymbol{\beta}_0 + \boldsymbol{\beta}_1 - p\boldsymbol{\beta}_1 - 2\gamma p\boldsymbol{\beta}_0 - \gamma\boldsymbol{\beta}_1 + 2\gamma p\boldsymbol{\beta}_1 + \gamma\boldsymbol{\beta}_0 & (30) \\
&= \mathbb{E}_{true} - 2\gamma p\boldsymbol{\beta}_0 - \gamma\boldsymbol{\beta}_1 + 2\gamma p\boldsymbol{\beta}_1 + \gamma\boldsymbol{\beta}_0 & (31) \\
&= \mathbb{E}_{true} - 2\gamma p\boldsymbol{\beta}_0 - \gamma\boldsymbol{\beta}_1 + 2\gamma p\boldsymbol{\beta}_1 + \gamma\boldsymbol{\beta}_0 - \gamma\boldsymbol{\beta}_1 + \gamma\boldsymbol{\beta}_1 & (32) \\
&= \mathbb{E}_{true} - 2\gamma p\boldsymbol{\beta}_0 - 2\gamma\boldsymbol{\beta}_1 + 2\gamma p\boldsymbol{\beta}_1 + \gamma\boldsymbol{\beta}_0 + \gamma\boldsymbol{\beta}_1 & (33) \\
&= \mathbb{E}_{true} - 2\gamma \left( p\boldsymbol{\beta}_0 + \boldsymbol{\beta}_1 - p\boldsymbol{\beta}_1 \right) + \gamma\boldsymbol{\beta}_1 + \gamma\boldsymbol{\beta}_0 & (34) \\
&= (1 - 2\gamma) \mathbb{E}_{true} + 2\gamma\mathbb{E}_{rand} & (35)
\end{aligned}
$$

## A.2 TYPE II NOISE

Type II noise implies class-dependent label noise. Thus, we define $\gamma_0$ and $\gamma_1$, the class-dependent noise rates. In addition, we define $\gamma^*$ as the average class-dependent noise rate and $\epsilon$ as the absolute distance of $\gamma_0$ and $\gamma_1$ from $\gamma^*$. Without loss of generality, we assume $\gamma_0 > \gamma_1$.

$$
\begin{aligned}
\gamma_0 &= p(\tilde{y}_i = 1 | y_i = 0) \\
\gamma_1 &= p(\tilde{y}_u = 0 | y_i = 1) \\
\gamma^* &= \frac{\gamma_0 + \gamma_1}{2} \\
\epsilon &= \frac{\gamma_0 - \gamma_1}{2}
\end{aligned}
$$

We may redefine $p(\tilde{y}_i = 0)$ and $p(\tilde{y}_i = 1)$ using these terms:

$$
\begin{aligned}
p(\tilde{y}_i = 0) &= ((1 - \gamma_0)\, p + \gamma_1\, (1 - p)) & (36) \\
&= ((1 - \gamma^* - \epsilon)\, p + (\gamma^* - \epsilon)\, (1 - p)) & (37) \\
&= (p - \gamma^* p - \epsilon p + \gamma^* - \gamma^* p - \epsilon + \epsilon p) & (38)
\end{aligned}
$$

$$
\begin{aligned}
p(\tilde{y}_i = 1) &= ((1 - \gamma_1)\, (1 - p) + \gamma_0 p) & (39) \\
&= ((1 - \gamma^* + \epsilon)\, (1 - p) + (\gamma^* + \epsilon)\, p) & (40) \\
&= (1 - \gamma^* + \epsilon - p + \gamma^* p - \epsilon p + \gamma^* p + \epsilon p) & (41)
\end{aligned}
$$

Plugging this into Equation 8,

$$
\begin{aligned}
\mathbb{E}_{(\mathbf{X}, \tilde{Y}) \sim \tilde{P}} \left[ \nabla_{\boldsymbol{\theta}} \hat{R}[\ell, \tilde{\mathcal{T}}] \right] &= (p - \gamma^* p - \epsilon p + \gamma^* - \gamma^* p - \epsilon + \epsilon p)\boldsymbol{\beta}_0 & (42) \\
&+ (1 - \gamma^* + \epsilon - p + \gamma^* p - \epsilon p + \gamma^* p + \epsilon p)\boldsymbol{\beta}_1 \\
&= (p - 2\gamma^* p + \gamma^* - \epsilon)\boldsymbol{\beta}_0 & (43) \\
&+ (1 - 2\gamma^* + \epsilon - p + 2\gamma^* p + \gamma^*)\boldsymbol{\beta}_1
\end{aligned}
$$

Similar to our approach with Type I noise, we aim to rephrase this equation in terms of $\mathbb{E}_{true}$ and $\mathbb{E}_{rand}$.

$$
\begin{aligned}
&= (1 - 2\gamma^*)\, p\boldsymbol{\beta}_0 + (\gamma^* - \epsilon)\, \boldsymbol{\beta}_0 + (1 - 2\gamma^*)\, (1 - p)\, \boldsymbol{\beta}_1 + (\epsilon + \gamma^*)\, \boldsymbol{\beta}_1 & (44) \\
&= (1 - 2\gamma^*)\, \mathbb{E}_{true} + (\gamma^* - \epsilon)\, \boldsymbol{\beta}_0 + (\epsilon + \gamma^*)\, \boldsymbol{\beta}_1 & (45) \\
&= (1 - 2\gamma^*)\, \mathbb{E}_{true} + 2\gamma^* \mathbb{E}_{rand} + \epsilon\, (\boldsymbol{\beta}_1 - \boldsymbol{\beta}_0) & (46)
\end{aligned}
$$

## A.3 TYPE III NOISE

In this section, we analyze a specific case of type III noise where label corruption is restricted to a given region of the feature space. As above, we want to study the effect on learning by determining how the true gradient is modified in the presence of noise. We assume a scalar input space and define the feature dependent noise as follows:

$$
p(\tilde{y}_i | x) = \left\{ \begin{array}{ll} \gamma & x \in [a, b] \\ 0 & x \notin [a, b] \end{array} \right\}
$$

The noise model above asserts that label noise depends only on the feature space and is label-agnostic - thus, examples from both class 1 and class 0 with $x \in [a, b]$ are equally likely to suffer from a label flip.

Revisiting

$$
\begin{aligned}
\mathbb{E}_{(\mathbf{X},\tilde{Y})\sim\tilde{P}}\left[\nabla_{\boldsymbol{\theta}}\hat{R}[\ell,\tilde{\mathcal{T}}]\right] &= \nabla_{\boldsymbol{\theta}}\left[\frac{1}{n}\sum_{i=1}^{n}\int p(\tilde{y}_i|x_i)p(x_i)\ell\left(\Phi(x_i,\boldsymbol{\theta}),\tilde{y}_i\right)\,dx_i\,d\tilde{y}_i\right], \\
&= \nabla_{\boldsymbol{\theta}}\left[\frac{1}{n}\sum_{i=1}^{n}\int_a^b p(\tilde{y}_i|x_i)p(x_i)\ell\left(\Phi(x_i,\boldsymbol{\theta}),\tilde{y}_i\right)\,dx_i\,d\tilde{y}_i\right] \\
&+ \nabla_{\boldsymbol{\theta}}\left[\frac{1}{n}\sum_{i=1}^{n}\int_{x\notin[a,b]} p(y_i|x_i)p(x_i)\ell\left(\Phi(x_i,\boldsymbol{\theta}),\tilde{y}_i\right)\,dx_i\,d\tilde{y}_i\right].
\end{aligned}
$$

Now we add and subtract the following term:

$$
\nabla_{\boldsymbol{\theta}}\left[\frac{1}{n}\sum_{i=1}^{n}\int_a^b p(y_i|x_i)p(x_i)\ell\left(\Phi(x_i,\boldsymbol{\theta}),\tilde{y}_i\right)\,dx_i\,d\tilde{y}_i\right] \tag{47}
$$

In order to obtain:

$$
\begin{aligned}
\mathbb{E}_{(\mathbf{X},\tilde{Y})\sim\tilde{P}}\left[\nabla_{\boldsymbol{\theta}}\hat{R}[\ell,\tilde{\mathcal{T}}]\right] &= \mathbb{E}_{true} + \nabla_{\boldsymbol{\theta}}\left[\frac{1}{n}\sum_{i=1}^{n}\int_a^b p(\tilde{y}_i|x_i)p(x_i)\ell\left(\Phi(x_i,\boldsymbol{\theta}),\tilde{y}_i\right)\,dx_i\,d\tilde{y}_i\right] \\
&- \nabla_{\boldsymbol{\theta}}\left[\frac{1}{n}\sum_{i=1}^{n}\int_a^b p(y_i|x_i)p(x_i)\ell\left(\Phi(x_i,\boldsymbol{\theta}),\tilde{y}_i\right)\,dx_i\,d\tilde{y}_i\right] \\
&= \mathbb{E}_{true} + \nabla_{\boldsymbol{\theta}}\left[\frac{1}{n}\sum_{i=1}^{n}\int_a^b \left(p(\tilde{y}_i|x_i)-p(y_i|x_i)\right)p(x_i)\ell\left(\Phi(x_i,\boldsymbol{\theta}),\tilde{y}_i\right)\,dx_i\,d\tilde{y}_i\right]
\end{aligned}
$$

Where we have used the fact that:

$$
\begin{aligned}
\mathbb{E}_{true} &= \nabla_{\boldsymbol{\theta}}\left[\frac{1}{n}\sum_{i=1}^{n}\int_a^b p(y_i|x_i)p(x_i)\ell\left(\Phi(x_i,\boldsymbol{\theta}),\tilde{y}_i\right)\,dx_i\,d\tilde{y}_i\right] \\
&+ \nabla_{\boldsymbol{\theta}}\left[\frac{1}{n}\sum_{i=1}^{n}\int_{x\notin[a,b]} p(y_i|x_i)p(x_i)\ell\left(\Phi(x_i,\boldsymbol{\theta}),\tilde{y}_i\right)\,dx_i\,d\tilde{y}_i\right]
\end{aligned}
$$

## B    ALGORITHM PSEUDO-CODE

In this section, we include a more detailed form of the algorithms proposed in the paper.

---

**Algorithm 1** Train energy-based autoencoder

---

**Require:** $\tilde{\mathcal{T}}$                  ▷ full training data
**Require:** $\mathscr{T}$              ▷ training data with known labels
**Require:** $\eta$                   ▷ learning rate
**Require:** Forward $(\boldsymbol{x}, \boldsymbol{\theta})$      ▷ autoencoder transformation that maps $\forall \boldsymbol{x} \to \hat{\boldsymbol{x}}$
**Require:** $E(\hat{\boldsymbol{x}})$        ▷ energy function, i.e. reconstruction loss
**Require:** $\mathcal{L}(\mathcal{B}, \mathcal{O}; \boldsymbol{\theta})$        ▷ objective function as defined in eq. 5
**Require:** GetBalancedBatch $(\mathscr{T})$      ▷ get mini-batch with an equal number
                    of instances from each class

1: **for** $i$ in $\{1,2,\ldots,m\}$ **do**         ▷ where $m$ is total number of batches
2:    $\mathcal{B}_i = \{\boldsymbol{x}_j, y_j\} \leftarrow$ GetBalancedBatch $(\mathscr{T})$
3:    $\mathcal{O}_i = \{\hat{\boldsymbol{x}}_j\} \leftarrow$ Forward $(\mathcal{B}_i, \boldsymbol{\theta}_i)$
4:    $\ell_i \leftarrow \mathcal{L}(\mathcal{B}_i, \mathcal{O}_i; \boldsymbol{\theta}_i)$         ▷ $\mathcal{L}(\mathcal{B}_i, \mathcal{O}_i; \boldsymbol{\theta}_i)$ is defined in Eq. 5
5:    $\nabla_{\boldsymbol{\theta}}(\ell_i) \leftarrow$ Backward $(\ell_i; \boldsymbol{\theta}_i)$
6:    $\boldsymbol{\theta}_{i+1} \leftarrow$ Step $(\boldsymbol{\theta}_i, \eta, \nabla_{\boldsymbol{\theta}}(\ell_i))$
7: **end for**

8: **Initialize:** $\mathcal{E} = \{\}$         ▷ empty dictionary that will store energies
9: **for** $\{\boldsymbol{x}_i, \tilde{y}_i\}$ in $\tilde{\mathcal{T}}$ **do**
10:    $\hat{\boldsymbol{x}}_i \leftarrow$ Forward $(\boldsymbol{x}_i; \boldsymbol{\theta}^\star)$      ▷ where $\theta^\star$ is the output of previous loop
11:    $e_i \leftarrow E(\hat{\boldsymbol{x}}_i)$
12:    $\mathcal{E}[i] \leftarrow e_i$          ▷ $\mathcal{E}[i]$ is the value of $\mathcal{E}$ at index $i$
13: **end for**

---

**Algorithm 2** Correct training data

---

**Require:** $\tilde{\mathcal{T}}$                  ▷ full training data
**Require:** $\mathscr{T}$              ▷ training data with known labels
**Require:** $\mathcal{E}$               ▷ energies from algorithm 1
**Require:** $E(\hat{\boldsymbol{x}})$        ▷ energy function, i.e. reconstruction loss
**Require:** GetNegativeSamples $(\mathscr{T})$       ▷ get all samples from class 0
**Require:** GetPositiveSamples $(\mathscr{T})$       ▷ get all samples from class 1

1: $\mathscr{T}_0 \leftarrow$ GetNegativeSamples $(\mathscr{T})$
2: $\mathscr{T}_1 \leftarrow$ GetPositiveSamples $(\mathscr{T})$
3: $\bar{E}_0 \leftarrow \frac{1}{|\mathscr{T}_0|} \sum_{\boldsymbol{x} \in \mathscr{T}_0} E(\boldsymbol{x})$        ▷ get mean w.r.t class 0
4: $\bar{E}_1 \leftarrow \frac{1}{|\mathscr{T}_1|} \sum_{\boldsymbol{x} \in \mathscr{T}_1} E(\boldsymbol{x})$        ▷ get mean w.r.t class 1
5: $\sigma_0 \leftarrow \left[ \sum_{\boldsymbol{x} \in \mathscr{T}_0} \frac{\left(E(\boldsymbol{x};\boldsymbol{\theta}) - \bar{E}_0\right)^2}{|\mathscr{T}_0| - 1} \right]^{\frac{1}{2}}$      ▷ get standard deviation w.r.t class 0
6: $\sigma_1 \leftarrow \left[ \sum_{\boldsymbol{x} \in \mathscr{T}_1} \frac{\left(E(\boldsymbol{x};\boldsymbol{\theta}) - \bar{E}_1\right)^2}{|\mathscr{T}_1| - 1} \right]^{\frac{1}{2}}$      ▷ get standard deviation w.r.t class 1
7: $\mathscr{T}^c \leftarrow \tilde{\mathcal{T}} \setminus \mathscr{T}$     ▷ i.e. $\mathscr{T}^c = \left\{ (\boldsymbol{x}_i, y_i) \in \tilde{\mathcal{T}} \,\|\, (\boldsymbol{x}_i, y_i) \notin \mathscr{T} \right\}$
8: **for** $\{\boldsymbol{x}_i, y_i\}$ in $\mathscr{T}^c$ **do**
9:    $e_i \leftarrow \mathcal{E}[i]$          ▷ get energy corresponding to $\boldsymbol{x}_i$
10:    **if** $y_i = 1$ **and** $e_i < \bar{E}_0 + \sigma_0$ **then**
11:      $y_i \leftarrow 0$
12:    **else if** $y_i = 0$ **and** $e_i > \bar{E}_1 - \sigma_0$ **then**
13:      $y_i \leftarrow 1$
14:    **end if**
15: **end for**

---

