# OpenReview forum: "An Energy-Based Framework for Arbitrary Label Noise Correction"
_ICLR.cc/2019/Conference_

### Official Review · AnonReviewer2 · 2018-11-01
**Good intuition with weak theoretical supports**

**Rating:** 5
**Confidence:** 5

**Review:**

This submission proposes an energy-based method to correct mislabelled examples. Intuitively, the authors claim that contradictions between energy and noisy labels can be used to identify label noise. To make the idea reliable, the authors propose to compute the energy by using learned (commonly shared) features. Experiment results look good. The presentation is also clear. My concerns are as follows:

(1) By learning discriminative features and then correcting the label noise, the authors have implicitly assumed that the label noise strongly dependent on the discriminative features. This assumption may be strong as most labels are provided according to the original instance (features). In the experiment section, it is very unclear about how the noise is generated, e.g., how to select x_i according to variance? How to set the threshold? What is "col" in Tables 1 and 2? What are the overall label noise rates? Note that if the threshold for the variance is large. It means that the noise is only added to the discriminative features, making the experiment too ad-hoc.

(2) The theory of why the residual energy can be used to identify label noise is elusive. How to set the threshold for identifying label noise with a theoretical guarantee is also unclear. Two recent papers on learning with instance-dependent label noise are surprisingly missed, e.g., Menon, Aditya Krishna, Brendan Van Rooyen, and Nagarajan Natarajan. "Learning from binary labels with instance-dependent corruption." arXiv preprint arXiv:1605.00751 (2016). and Cheng, Jiacheng, et al. "Learning with Bounded Instance-and Label-dependent Label Noise." arXiv preprint arXiv:1709.03768 (2017). The latter one proposes algorithms to identify label noise with theoretical guarantees. The authors should compare the proposed method with them.

(3) There are methods provided for choose the values of hyperparameters. Most of them are empirically set, which is not convincing.

(4) The authors reported that with discriminative features learned by employing noisy data, the proposed method also provide good performance. It would be interesting to see how corrected labels will recursively help better learn the discriminative features. Illustrating figures are preferred.

---

### Official Review · AnonReviewer3 · 2018-11-05
**Need improvements**

**Rating:** 5
**Confidence:** 4

**Review:**

Need improvements

[Summary]

This paper addresses the problem of correcting noisy labels for binary classification. It assumes the exists of fully clean data, trains an energy-based autoencoder using contrastive learning objective, and use the estimated energy to determine if a training label is corrupted or not.

[Pros]

1.	The paper summarizes different types of label noise in a sensible way. And, it is reasonable to bootstrap the learning process with a small fully clean dataset.
2.	The proposed method shows encouraging results under controlled noise.

[Cons]

1.	It is not well-motivated why a contrastive objective or an energy-based autoencoder can be a good solution for label correction. In particular, the connections are not established between the discriminative feature learned by an energy-based model and the label correctness. The proposed method looks more like a binary classifier with a better-regularized structure, but still, it is unclear why an energy-based autoencoder is a good choice.
2.	The proposed method is limited to binary classification, and there is no obvious way to extend it to multiple classes.
3.	The experiments are on toy/small-scale datasets with controlled label noise (but the way to control the noise is not clear). To show the effectiveness of the proposed methods, experiments need to be done on larger-scale datasets with truly realistic unknown noise, Establishing state-of-the-art classification accuracy using a large-scale dataset with noisy labels can serve as strong support for this paper.

---

> ### Author Response · Authors · 2018-11-19
> **Authors' response**
>
> The authors thank the reviewer for recognizing that the presented results are encouraging and that the paper summarizes label noise in an acceptable way. We also agree with the reviewer that appropriate assumptions are made in setting up the problem of label noise correction.
>
> To address some of the authors remarks, we would like to highlight that the existing method can use a small dataset that has clean labels to facilitate the label correction process. However, as shown in tables 1, 2 and 3 (see column AE learned), one does not need to start with clean labels. Using the method of Ding et al. (2018), clean labels can be inferred on a subset of the data before bias correction step is applied.
>
> Reviewer: “It is not well-motivated why a contrastive objective or an energy-based autoencoder can be a good solution for label correction. In particular, the connections are not established between the discriminative feature learned by an energy-based model and the label correctness. The proposed method looks more like a binary classifier with a better-regularized structure, but still, it is unclear why an energy-based autoencoder is a good choice.”
>
> The authors agree that this should be explained more clearly in the paper (appropriate changes to follow). The following illustrates the motivation of the contrastive loss and the energy interpretation:
> Consider the case when noise is only in one class and the other class is noise free. Then, one could train an autoencoder (with standard reconstruction loss) to become accurate at reconstructing samples from the noise free class. Once the autoencoder learned weights have been set, it can be used to process the noisy class (where some of the labels are incorrect). Then, one would expect that the miss-labelled instances will have low reconstruction loss (i.e. low energy), which has been inferred using an unsupervised approach (i.e. autoencoder with reconstruction loss) instead of taking a fully supervised approach. This is because the features of these mis-labelled instances actually correspond to the features that were used to train the original autoencoder.  Our contrastive loss guarantees that the loss will be large for samples that are not from the noise free class in the above example. That is, the negative sign in front of the second term in equation (5) ensures that the energy will be large for the second class. As such the first term in equation (5) should correspond to the class that has the most clean samples out of the two classes (as mentioned in ). In summary, the connection to label noise is that the energy value indicates whether ambiguous samples (i.e. samples with unreliable labels) have features that are similar to the class corresponding to term 1 of equation (5).
>
> To address another one of the reviewer’s comments, we would like to mention that our approach may extend to multiclass scenarios by constructing a one-vs-all scheme. In this paper, we aim to augment literature for label correction in binary classification.
>
> Reviewer: "The experiments are on toy/small-scale datasets with controlled label noise (but the way to control the noise is not clear). To show the effectiveness of the proposed methods, experiments need to be done on larger-scale datasets with truly realistic unknown noise, Establishing state-of-the-art classification accuracy using a large-scale dataset with noisy labels can serve as strong support for this paper."
>
> Towards this end, we include the Arrhythmia dataset results (Table 1, last row) and the MNIST dataset with label noise as studied by Ren et al. (2018). The Arrhythmia dataset consists of 50000 samples and the problem posed - of algorithmically assigned labels - remains relevant to real world tasks such as web annotation. This dataset contains real label noise as discussed in the paper. Next, we compare our method with another state-of-the-art noise robust algorithm (Ren et al. (2018)) on a relatively large dataset: MNIST.

---

### Official Review · AnonReviewer1 · 2018-11-08
**Need more theoretical guarantee**

**Rating:** 5
**Confidence:** 4

**Review:**

This paper addresses Type III label noise correction problem in which the labeling noise depends on the features. They assume that we can obtain a small amount of cleanly labeled data, and use an energy-based semi-supervised learning approach to bootstrap the relabeling process.

Pros:
- Problem is well-motivated with a reasonably good overview of this research area.
- Paper is generally well-written with enough details to follow and good experimental result discussion.

Cons:
- The energy-based approach based on contrastive divergence is pretty straightforwardly defined, but it will make the paper much stronger if the authors can have more analysis on this approach and/or provide theoretical guarantee on generalization.
- It is not obvious to me how to extend the proposed approach to multi-class problems.
- It will be beneficial to test the approach on more real-world problems on top of the toy-data-alike binary classification problems.

Minor clarification questions:
- What amount of cleanly labeled data is sufficiently required for the proposed approach to work? The authors have some pre-selected percentage in experiments but it is non-trivial to establish that for different applications.
- Related to the previous comment, how much clean data were used in AE (known) columns in all experiments?
- In Fig 2 between the two subgraphs, why is the left one showing positive thetas while the right one showing negative thetas?
- Were the hyperparameters a & b chosen from cross-validation or from std of E terms in all experiment results?
- In Table 1, for Breast Cancer dataset, how can AE (known) be better than the upper-bound LR-C?
- It would be good to vary the noise parameters and show how robust the proposed approach is in dealing with different levels of noise.

---

> ### Author Response · Authors · 2018-11-20
> **Authors' response**
>
> The authors thank the reviewer for recognizing that the problem under consideration is well motivated and that the paper is well-written and presents good experimental results.
>
> Reviewer: “This paper addresses Type III label noise correction problem in which the labeling noise depends on the features. They assume that we can obtain a small amount of cleanly labeled data, and use an energy-based semi-supervised learning approach to bootstrap the labeling process.”
>
> This point merits further clarification in the paper. The proposed method does not rely on a small amount of cleanly labeled data, but rather accommodates for the existence of this set. The proposed method includes a step to infer which points fall into this, in Section 4’s Step 1.
>
> To answer the reviewer’s concern about extending this to multi-class classification, we would like to highlight that our approach may extend to multi-class scenarios by constructing a one-vs-all scheme. Nevertheless, the primary focus of this paper was to study the problem of label noise in the case of binary classes.
>
> Reviewer: “It will be beneficial to test the approach on more real-world problems on top of the toy-data-alike binary classification problems.”
>
> Towards this end, we include the Arrhythmia dataset results (Table 1, last row) and the MNIST dataset with label noise as studied by Ren et al. (2018). The Arrhythmia dataset consists of 50000 samples and the problem posed - of algorithmically assigned labels - remains relevant to real world tasks such as web annotation. This dataset contains real label noise as discussed in the paper. Next, we compare our method with another state-of-the-art noise robust algorithm (Ren et al. (2018)) on a relatively large dataset: MNIST.
>
> Reviewer: “What amount of cleanly labeled data is sufficiently required for the proposed approach to work? The authors have some pre-selected percentage in experiments but it is non-trivial to establish that for different applications.”
>
> Using our inference procedure to identify clean data, we identify 10% of the dataset.
>
> Reviewer: “In Fig 2 between the two subgraphs, why is the left one showing positive thetas while the right one showing negative thetas?”
>
> The disparity in thetas explored between the two graphs highlight the effect of feature dependent label noise, in comparison to random label noise. On the left, the resulting parameters lie close to the true (gold) trajectory. On the right, the model follows a visibly different path. The distinction between position and negative thetas is an artifact of the simulated separation problem and the takeaway is the difference in behavior.
>
> Reviewer: “Were the hyperparameters a & b chosen from cross-validation or from std of E terms in all experiment results?”
>
> Hyperparameters a and b are selected using the heuristic outlined in Appendix B (Algorithm 2). That is, we set a and b to be the standard deviation of the energy for class 0 and class 1 respectively.
>
> Reviewer: “In Table 1, for Breast Cancer dataset, how can AE (known) be better than the upper-bound LR-C?”
>
> This can occur for a couple reasons. It could be that training on the smaller, higher confidence labels produces a more generalizable classifier, in the instance of certain test sets. We see that the standard deviations accompanying these measurements (.02 for the AE (known) performance and .01 for LR-C) demonstrate that the performance of these methods does that differ significantly.
>
> Reviewer: “It would be good to vary the noise parameters and show how robust the proposed approach is in dealing with different levels of noise.”
>
> Our experimentation with different noise parameters may be occluded by our presentation - we experiment with different feature-dependent noise models. In each of the tables, the ‘col’ variable represents the feature the label noise is dependent on. Table 1 represents experiments under the linear feature dependent noise model. Table 2 represents experiments under the quadratic feature dependent noise model. We vary the noise parameters as indicated on the second column of these tables. The different noise models and parameters tested in these experiments merit further explanation.
>
> The authors agree with the reviewer that this would be an interesting experiment, however, due to space restrictions, the authors can use the existing experiments to comment on the general expected trend resulting from increasing noise. This will be added to the next version of the paper.

---

### Meta-Review · Area_Chair1 · 2018-12-14
**not enough theoretical guarantees, evaluations are insufficient**

**Confidence:** 4
**Recommendation:** Reject

**Metareview:**

The authors present an algorithm for label noise correction when the label error is a function of the input features.

Strengths
- Well motivated problem and a well written paper.

Weaknesses
- The reviewers raised concerns about theoretical guarantees on generalization; it is not clear why energy based auto-encoder / contrastive divergence would be a good measure of label accuracy especially when the feature distribution has high variance, and when there are not enough clean examples to model this distribution correctly.
- Evaluations are all on toy-like tasks with small training sets, which makes it harder to gauge how well the techniques work for real-world tasks.
- It’s not clear how well the algorithm can be extended to multi-class problems. The authors suggested 1-vs-all, but have no experiments or results to support the claim.

The authors tried to address some of the concerns raised by the reviewers in the rebuttal, e.g., how to address unavailability of correctly labeled data to train an auto-encoder. But other concerns remain. Therefore, the recommendation is to reject the paper.